# Depression Earlier on in Life Predicts Frailty at 50 Years: Evidence from the 1958 British Birth Cohort Study

**DOI:** 10.3390/jcm12175568

**Published:** 2023-08-26

**Authors:** Paul Watts, Mukil Menon, Gopalakrishnan Netuveli

**Affiliations:** 1School of Health, Sport and Bioscience, University of East London, London E16 2RD, UK; p.n.watts@uel.ac.uk; 2Koniver Wellness UK, London SW1H 0BL, UK; 3Institute for Connected Communities, University of East London, London E16 2RD, UK

**Keywords:** frailty, depression, birth cohort, life course

## Abstract

Frailty and depression in older ages have a bidirectional relationship, sharing some symptoms and characteristics. Most evidence for this has come from cross-sectional studies, or longitudinal studies with limited follow-up periods. We used data from the National Child Development Study (1958 Birth Cohort) to investigate the relationship between depression and early-onset frailty using a life course perspective. The primary outcome was frailty based on a 30-item inventory of physical health conditions, activities of daily living and cognitive function at 50 years. The main exposure was depression (based on a nine-item Malaise score ≥ 4) measured at 23, 33 and 42 years. We investigated this relationship using multiple logistic regression models adjusted for socio-demographic factors, early life circumstances and health behaviours. In fully adjusted models, when modelled separately, depression at each timepoint was associated with around twice the odds of frailty. An accumulated depression score showed increases in the odds of frailty with each unit increase (once: OR 1.92, 95%CI 1.65, 2.23; twice OR 2.33, 95%CI 1.85, 2.94; thrice: OR 2.95, 95%CI 2.11, 4.11). The public health significance of this finding is that it shows the potential to reduce the physical burden of disease later in life by paying attention to mental health at younger ages.

## 1. Introduction

Frailty and depression are two health outcomes that, while frequently experienced by older adults, can also manifest at other points throughout the life course [1,2]. Both outcomes can have a profound impact on quality of life, independence, and healthcare utilisation throughout the ageing process [3,4,5,6]. Studies have indicated that the relationship between frailty and depression in older age is likely to be bidirectional, suggesting that each condition can influence the onset and progression of the other [7]. Moreover, frailty and depression have several common risk factors, such as chronic diseases and social isolation, share similar symptoms, such as fatigue and exhaustion [8] and may have common pathophysiological mechanisms, such as elevated diurnal cortisol [9].

Two commonly used measures of frailty are Fried’s Frailty Phenotype and Rockwood’s Frailty Index. Fried’s approach defines frailty as the presence of three or more of the following five conditions: unintentional weight loss, weakness, exhaustion, slow walking speed, and low physical activity [10]. On the other hand, Rockwood’s index encompasses a minimum of 30 items representing accumulated deficits, including cognitive tests, self-reported daily activity challenges, mental and physical health status, and specific disease presence [11]. In epidemiological studies, depression is often measured using standardised self-report surveys or structured clinical interviews [12,13]. Both of these approaches involve eliciting responses to questions to determine if a person meets the specific criteria for a disorder such as depression or can be classified as showing signs or symptoms of depression. Some components of these approaches used to define and measure frailty and depression may overlap, not only conceptually but also in the use of specific survey items. Specifically, certain survey items might appear on frailty indices, be used to measure one of Fried’s frailty criteria, and also be part of a scale or diagnostic tool used to measure depression. An example of a survey item that may be used for all of these purposes is the question “How often do you feel tired or have little energy?” [14].

The shared characteristics of frailty and depression, and similarities among approaches to measuring them, can complicate studies aiming to investigate the intricate relationship between these outcomes [8]. The current body of evidence suggests a bidirectional relationship between these two conditions in older age, characterised by shared symptoms and overlapping characteristics [15,16]. However, the majority of this evidence is derived from cross-sectional studies which offer a limited perspective on the temporal dynamics and causal pathways involved in this relationship [7,17]. Longitudinal studies are instrumental in elucidating whether the onset of frailty and depression are concurrent or if there is evidence that one condition generally precedes the other. However, the existing longitudinal studies have primarily focused on tracking adults already in their later years, with both exposure and outcome measures being assessed in old age [17]. Moreover, these longitudinal studies often have relatively short follow-up periods, which may not fully capture the long-term trajectories of frailty and depression [7]. This limitation is particularly pertinent given the chronic and progressive nature of both conditions, which may evolve over extended periods. 

A notable gap in the current literature is the absence of life course studies investigating the relationship between depression and frailty measured at different life stages. The current evidence base is dominated by studies that are limited to older adults, typically those over 65 years of age [16]. This leaves a need for studies investigating earlier-onset frailty and its potential association with depression earlier in life. An understanding of the factors contributing to earlier-onset physical frailty can inform opportunities for early intervention and contribute to an understanding of the natural history of frailty [1]. This is particularly important in relation to potentially modifiable risk factors such as depression. The ability to identify and modify these factors earlier in the life course could potentially delay or even prevent the onset of frailty. Indeed, previous research has indicated a higher prevalence of early-onset frailty in individuals with other chronic conditions, such as diabesity—the coexistence of diabetes and obesity. Understanding the points of frailty onset and identifying opportunities for early intervention is vital for the early detection of individuals at risk and for intervening on the first impacted components, where the likelihood of reversal might be highest [1]. 

Life course theories in epidemiology provide a framework for understanding how experiences throughout an individual’s lifespan can shape health outcomes, such as frailty and depression [18]. Two key theories are particularly relevant: (i) The ‘accumulation model’ posits that the impact of adverse experiences, such as chronic stress or trauma, can accumulate over time, potentially leading to conditions such as frailty [19]. The longer, more frequent, and more severe these exposures are, the greater the potential damage to an individual’s mental and physical health. (ii) The ‘critical period’ model suggests that there are specific ‘windows of time’, such as early childhood, where exposures to certain risk factors can have a profound impact on future health [20]. For instance, early life stress or trauma could increase the risk of developing depression or frailty later in life.

These theories underscore the importance of investigating life course factors that may be a common cause of depression and frailty, and therefore may be confounders of the relationship in epidemiological studies. Early life stress, which encompasses experiences like family disruption, poverty, or unsafe living conditions, has been shown to be associated with health issues including frailty in later life [21]. This relationship is thought to stem from the ‘biological embedding’ of stress, where chronic early life stress can alter the body’s stress response systems, influencing health outcomes in adulthood [22]. Lower socioeconomic status at different life stages, measured by educational level, occupational status and financial status, has been linked to higher rates of depression [23] and frailty in later life [24]. This association is believed to be due to a range of factors, including increased exposure to stressful events, limited access to healthcare, and a higher likelihood of engaging in behaviours detrimental to health.

The identified need for longitudinal studies, covering periods of the life course prior to older age, able to identify frailty at early-onset, and accounting for relevant confounders at different stages of life, presents the following research questions: (a)Is depression measured at ages earlier in adulthood associated with frailty measured at age 50?;(b)Do these associations remain after adjusting for socio-demographic covariates measured at age 50, early life circumstances, and health risk behaviours?

## 2. Materials and Methods

### 2.1. National Child Development Study Data

The National Child Development Study (NCDS) is a birth cohort study of children born in Scotland, Wales and England in one week of 1958 [25] with regular follow-up survey sweeps [26]. These survey sweeps include information on educational and social development in early life and health outcomes, health behaviours and socio-economic circumstances throughout the life course. The analyses reported here utilise NCDS life course data on early life social and economic circumstances, early adulthood health behaviours and socio-economic circumstances, health outcomes, activities of daily living and cognitive function at age 50. A cohort of 17,415 participants entered the NCDS at birth in 1958. Of the total, 9789 participants who completed the survey at sweep 8 (age 50) form the analytic sample for the present study.

### 2.2. Outcome Measure: Frailty at Age 50

A 30-item frailty index was created following the methods described by Searle and colleagues [27]. The index included items covering physical health conditions; self-rated health; activities of daily living (ADLs); body mass index and general cognition. We excluded from the index any items that overlap substantively with our main exposure variable such as symptoms of depression or exhaustion [28]. Each item was scaled to a value between 0 and 1. For binary items like specific health conditions and difficulty climbing one flight of stairs, a deficit was scored 1 and no deficit 0. For Likert scale items like self-reported health status, the scores were: Poor = 1; Fair = 0.75; Good = 0.5; Very good = 0.25; Excellent = 0. For continuous variables like scores on a word recall test, the total score was transformed into quintiles: 5th Quintile = 1; 4th Quintile = 0.75; 3rd Quintile = 0.5; 2nd Quintile = 0.25; 1st Quintile = 0. See Appendix A for full details of items included in the frailty index. Frailty scores ranging between 0 and 1 were calculated for each participant by summing the completed item scores and then dividing the sum by the number of answered items [11]; those with a score of 0.25 or above were defined as frail. This cutoff point of 0.25 is most commonly used in previous research and has been shown to represent a frail state using multiple methods [29]. 

### 2.3. Main Exposure: Depression at Ages 23, 33 and 42

The main exposure in this study was depression measured with the 9-item Malaise Inventory [30] at ages 23, 33 and 42 years, used separately and also as an accumulation score representing the number of times a participant was classified as having depressive symptoms (with a possible range of 0 to 3 times). The Malaise Inventory score is calculated by summing the positive responses to nine binary items such as “Do you often feel miserable or depressed?” and “Do you often get worried about things?”, where the responses were ‘Yes’ = 1 and ‘No’ = 0, creating a possible score range of 0 to 9. Following previous research, we dichotomised this score with ≥4 as the cutoff point, representing an indication of depressive symptoms [31]. We chose the nine-item version of the Malaise Inventory as it has been applied consistently across NCDS survey sweeps and previous studies have shown that this measure has acceptable measurement invariance properties [31]. Furthermore, the nine-item version does not contain any items from the full 24-item version that represent physical symptoms, therefore avoiding overlap with our outcome measure [28].

### 2.4. Covariates

We selected a range of relevant covariates at different life course stages based on their theorised relationship with both the exposure and the outcome and evidence from previous studies indicating their likely association with the exposure and outcome. Socio-demographic covariates measured at age 50 were: (i) Sex (Male/Female); (ii) Marital status (Married or Civil partnership/Single, Never married or Civil partnership/Separated or Divorced/Widowed); (iii) Employment status at age 50 (Employed = 1/Not employed = 0); (iv) Subjective financial position at age 50 (Likert scale: ‘Living comfortably’ = 0/‘Doing all right’ = 1/‘Just about getting by’ = 2/‘Finding it quite difficult’ = 3/‘Finding it very difficult’ = 4).

Covariates measured during early life were: (i) Socio-economic circumstances at birth using a binary measure of the social class of the father’s occupation, as allocated (manual or non-manual) by the Registrar General’s classification. (ii) An accumulated measure of childhood disadvantage [32], defined as a count of exposures to the following disadvantages: (a) at birth, if the NCDS participant’s father was employed in an occupation allocated to Registrar General’s social class IV or V; (b) at age 7, the NCDS participants’ parents reported having financial difficulties; (c) at age 11, the NCDS participants received free school meals; (d) at age 16, the NCDS participants’ parents self-reported financial difficulties. (iii) At age 7, discord between the NCDS participant’s parents was reported by a health visitor (Yes = 1/No = 0). (iv) At age 7, a parent of the NCDS participant had died or the parents had separated (Yes = 1/No = 0). (v) The Bristol Social Adjustment Guide (BSAG) total score was used to evaluate social and emotional competencies at ages 7 and 11. The full BSAG comprises 146 distinct ‘behavioural’ components, categorised into 12 separate ‘syndromes’: Anxiety for acceptance by adults; Anxiety for acceptance by children; Restlessness; ‘Inconsequential’ behaviour; Withdrawal; Depression; Hostility towards adults; Hostility towards children; ‘Writing off’ of adults and adult standards; Unforthcomingness; Miscellaneous nervous symptoms; Miscellaneous symptoms. Teachers administering the BSAG underlined items that they believed to accurately depict the child’s characteristics, each underlined item was assigned a score of 1 [33]. To mitigate measurement error and the potential impact of ‘shocks’, we computed the mean of the scores from ages 7 and 11 [34]. Mean BSAG scores were then categorised into quartiles with the 4th quartile viewed as most maladjusted [35]. 

Health behaviours measured at age 16 and 23 were: (i) Physical activity at age 16. Participants responded to items asking how frequently they participated in (a) outdoor games and sports; (b) indoor games and sports; (c) swimming; (d) dancing. Responses were coded as often = 2/sometimes = 1/never or hardly ever = 0. These responses were summed to create a score between 0 and 9 and a cutoff point of ≥4 was used to classify participants as active/inactive [36]. (ii) Smoking at age 23 was defined as a binary variable where smokers and past smokers = 1/non-smokers = 0. (iii) Alcohol consumption at age 23 was defined using responses to questions about consumption of various alcoholic drinks (beer, wine, spirits, vermouth or sherry). Responses were translated into standard UK units of alcohol and summed across items. A cutoff point of >14 weekly units was used to indicate consumption above recommended levels [37].

### 2.5. Data Analysis

Associations between frailty at age 50 and depression at ages 23, 33, and 42 were estimated using multiple logistic regression models. Due to expected autocorrelation between depression measured in the same individuals at different timepoints, we fitted models separately for depression measured at ages 23, 33, and 42 and a further model was fitted using the accumulated depression score. Models were progressively adjusted for sociodemographic covariates at age 50, early life circumstances, and health behaviours at ages 16 and 23. Missing data were accounted for using multiple imputation chained equations (20 repetitions) following recommendations provided by the NCDS missing data user guidance [38]. To strengthen assumptions about the direction of causation, sensitivity analyses were conducted after excluding participants with a physical handicap or disabling condition at age 7 or a longstanding illness or disability at age 23. All analyses were conducted in Stata version 15.1.

## 3. Results

### 3.1. Descriptive Statistics

At age 50 (sweep 8), 9789 participants completed the NCDS survey, of which 51% were female. Overall, around thirty percent (30.1%) of the sample had a frailty index score of 0.25 or above and were therefore classified as frail. There were very similar proportions of frail women (30.2%) and men (30.0%). At age 23, around nine percent (8.9%) of the sample were classified as indicating depressive symptoms according to their Malaise Inventory score. This proportion dropped to 7.8% at age 33, but increased to 13.0% by age 42. At all three survey sweeps, the proportion of women indicating depressive symptoms was higher than men. Descriptive data on all participant characteristics, levels of frailty and depression are shown in Table 1 for both complete cases and proportions estimated using the imputed dataset.

### 3.2. Associations between Depression in Adulthood and Frailty at Age 50

Associations between depression in adulthood and frailty at age 50 are presented in Figure 1. When modelled separately, without adjustment for covariates, depression was significantly associated with frailty at age 23 (OR 2.65, 95%CI 2.26, 3.12) age 33 (OR 3.28, 95%CI 2.73, 3.93) and age 42 (OR 3.09, 95%CI 2.72, 3.50). In models fully adjusted for all covariates, the strength of these associations was attenuated, but remained statistically significant at age 23 (OR 1.76, 95%CI 1.47, 2.10), age 33 (OR 2.11, 95%CI 1.72, 2.60) and age 42 (OR 2.15, 95%CI 1.87, 2.48). Using the accumulated depression score, without adjusting for covariates, the odds of frailty increased significantly with each unit increase (once: OR 2.43, 95%CI 2.11, 2.79; twice: OR 3.57, 95%CI 2.89, 4.41; thrice: OR 5.83, 95%CI 4.32, 7.88). This association remained statistically significant in a model fully adjusted for all covariates (once: OR 1.92, 95%CI 1.65, 2.23; twice: OR 2.33, 95%CI 1.85, 2.94; thrice: OR 2.95, 95%CI 2.11, 4.11). All associations remained with minimal changes in effect size in sensitivity analyses conducted after excluding participants with a physical handicap or disabling condition at age 7 or a longstanding illness or disability at age 23. 

### 3.3. Covariates

In models adjusted for all covariates, socio-demographic factors at age 50 were strongly associated with frailty (see Table 2). The odds of frailty were significantly increased for participants who reported not being employed at age 50, for those who had lower levels of education, and those reporting more difficulty with their current financial position. Marital status at age 50 was not associated with frailty. Covariates measured during early life that were associated with significantly increased odds of frailty were father’s occupational social class reported as ‘manual’, having parents who had divorced, separated or died by age 7, and having a mean BSAG score in the 3rd or 4th quartile. Smoking at age 23 was associated with a small increase in the odds of frailty, while alcohol consumption above the recommended 14 units per week was associated with a small decrease in the odds of frailty at age 50. All modelled associations between covariates and frailty are presented in Table 2.

## 4. Discussion

Our study provides evidence for a significant association between depression at earlier life stages and frailty at age 50, as measured with a 30-item inventory. This relationship was observed consistently across different timepoints and remained significant after adjusting for a range of life course covariates. When examined separately, depression at ages 23, 33, and 42 was associated with approximately double the odds of frailty. Furthermore, in the accumulated model, participants who exhibited symptoms of depression at all three timepoints had almost triple the odds of frailty compared to those without symptoms at these timepoints, suggesting a cumulative effect of depression over time on the risk of frailty. These findings add to evidence from previous studies reporting associations between depression and frailty at later stages of the life course [7,16] by demonstrating that repeated episodes or chronic depression at earlier life stages may have a cumulative impact on the risk of frailty. This aligns with the life course ‘accumulation model’ [19] and underscores the importance of early and ongoing management of depression to potentially mitigate this risk.

Socio-demographic factors at age 50 were also strongly associated with frailty. Not being employed at age 50, having lower levels of education, and experiencing financial difficulties were all significantly associated with increased odds of frailty. This suggests that socio-economic factors in mid-life can have a substantial impact on physical health outcomes, potentially through mechanisms such as stress, and health risk behaviours [39]. Early life covariates that were significantly associated with increased odds of frailty included having a father with a manual occupation, experiencing parental divorce, separation, or death by age 7, and having a mean BSAG score in the 3rd or 4th quartile. These findings highlight the potential long-term impact of early life stressors and socio-economic conditions on frailty as have been reported in recent studies [21] and can be explained with the ‘critical period’ life course model [19].

The mechanisms and pathways through which depression impacts frailty are likely to be complex and multifaceted. Depression may increase the risk of frailty through its influence on health risk behaviours, such as physical inactivity, poor diet, and substance use, which are known risk factors for frailty [40]. Future research involving mediation analysis could help to elucidate the role of these behaviours in the relationship between depression and frailty. In addition to behavioural pathways, biological mechanisms may also play a role. Depression has been shown to be associated with chronic inflammation and hormonal imbalances, which could contribute to the development and progression of frailty [17]. Chronic inflammation can lead to muscle weakness and fatigue, which are key components of frailty, while hormonal imbalances, such as elevated cortisol levels, can affect metabolism and immune function, potentially exacerbating frailty [9]. Finally, the role of anti-depressants in this relationship warrants further investigation. Anti-depressants can help to manage the symptoms of depression, potentially reducing its impact on health behaviours and biological systems. However, their role in the risk of frailty is not well understood and there is emerging evidence to suggest potentially adverse effects of overprescribing antidepressants, which may offer minimal or no beneficial effects, and have important implications for the treatment of frailty in older adults [41]. 

The findings of our study highlight the importance of early identification and intervention for mental health issues, such as depression, as a potential strategy to reduce the risk of frailty in later life. This has several implications for policy and practice, such as integrating mental health screenings into routine healthcare to facilitate early detection of depression and improve access to appropriate interventions [42]. This could be particularly relevant for primary care settings, where most people have their first contact with the health system [43]. Implementing interventions aimed at promoting mental well-being from a young age could help to prevent the onset of depression, and potentially reduce the risk of subsequent frailty. This could involve school-based mental health promotion programs, public health campaigns aimed at raising awareness about mental health, and policies aimed at reducing risk factors for depression, such as poverty and social inequality [44]. Finally, our findings highlight the importance of considering mental health in the management of frailty. This could involve training healthcare professionals to recognise and manage mental health issues in people with frailty, and developing integrated care pathways that address both the physical and mental health needs of this population [45].

One of the primary strengths of our study lies in the use of longitudinal life course data. This approach has enabled us to examine the relationship between depression, measured at multiple timepoints, and early-onset frailty while adjusting for a range of relevant covariates measured from birth to mid-life. However, our study also has some limitations. We have created a frailty index following established methods and using a recommended minimum number of indicators covering a range of domains [11,27]. However, the validity of this frailty index is unknown and it has not been used in any previous study using NCDS data. The proportion of participants identified as frail in our study (30.1%) was slightly higher than the average prevalence of frailty measured using similar indices in studies of participants aged 50 to 59 (23%), but below the average rates of pre-frailty in this age group (41%) [46]. The frailty outcome we have used was associated with socio-economic factors such as education, employment status and financial position in congruence with many previous studies, but we do not currently have data available to validate the index against an alternative frailty measure such as one based on Freid’s criteria [10], or to examine its association with outcomes such as hospitalisation or mortality [47]. Similarly, further research, using datasets with alternative measures of depression available will be important in identifying whether the associations we have identified are consistent when using different measurement tools. 

Our study highlights several areas for future research to further the understanding of the relationship between depression and frailty. More research is needed to elucidate the mechanisms linking depression and frailty. This could involve exploring both behavioural and biological pathways, such as the impact of depression on health risk behaviours and the role of chronic inflammation and hormonal imbalances in relation to subsequent frailty. Longitudinal life course studies with longer follow-up periods could provide more insights into the long-term effects of depression on frailty beyond 50 years old. Future research could also explore how other social determinants, such as social support, neighbourhood environments, and access to healthcare, influence the relationship between depression and frailty. In addition to further observational studies, future research could also explore the potential role of interventions, such as those that act on health risk behaviours, or social isolation in mitigating the risk of frailty in individuals with depression earlier in life to inform a more integrated approach to the management of depression and frailty.

## 5. Conclusions 

In conclusion, our study finds a strong association between early-life depression and frailty at age 50, highlighting the cumulative effects of depression over time. Socio-demographic factors at mid-life and early-life stressors were other factors identified for their independent and significant roles in frailty onset. These findings build upon previous research on the association between depression and frailty, underscoring the importance of early detection and intervention for depression as a potential preventive measure. The intricate relationships between depression, health behaviours, and biological markers necessitate further exploration.

## Figures and Tables

**Figure 1 jcm-12-05568-f001:**
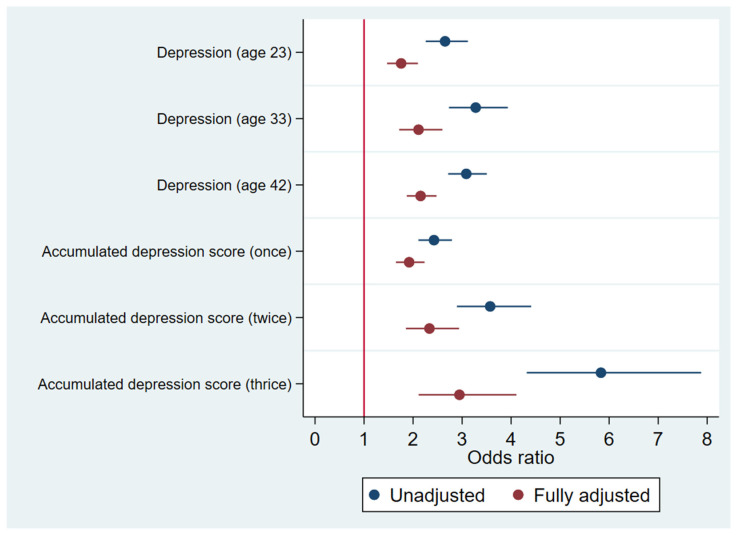
Associations between depression earlier in life and frailty at age 50.

**Table 1 jcm-12-05568-t001:** Sample statistics for all variables (complete cases and imputed dataset).

	All					Women					Men				
	Complete Cases		Imputed Dataset			Complete Cases		Imputed Dataset			Complete Cases		Imputed Dataset		
	N	%	%	LCI	UCI	N	%	%	LCI	UCI	N	%	%	LCI	UCI
**All**	9789	100.0				4968	50.8				4821	49.3			
**Frailty at Age 50**															
Non-frail (index score <0.25)	6814	69.6	69.9	69.0	70.8	3456	69.6	69.8	68.5	71.1	3358	69.7	70.0	68.7	71.3
Frail (index score ≥0.25)	2905	29.7	30.1	29.2	31.0	1485	29.9	30.2	28.9	31.5	1420	29.5	30.0	28.7	31.3
Missing	70	0.7				27	0.5				43	0.9			
**Depression at Age 23**															
Malaise inventory score <4	7551	77.1	91.1	90.5	91.7	3713	74.7	86.9	85.9	87.9	3838	79.6	95.5	94.8	96.1
Malaise inventory score ≥4	684	7.0	8.9	8.3	9.5	518	10.4	13.1	12.1	14.1	166	3.4	4.5	3.9	5.2
Missing	1554	15.9				737	14.8				817	17.0			
**Depression at Age 33**															
Malaise inventory score <4	7825	79.9	92.2	89.9	94.5	3943	79.4	89.5	86.9	92.2	3882	80.5	95.0	92.9	97.1
Malaise inventory score ≥4	571	5.8	7.8	5.5	10.1	402	8.1	10.5	7.8	13.1	169	3.5	5.0	2.9	7.1
Missing	1393	14.2				623	12.5				770	16.0			
**Depression at Age 42**															
Malaise inventory score <4	7874	80.4	87.0	86.3	87.7	3932	79.2	84.0	83.0	85.1	3942	81.8	90.0	89.1	90.9
Malaise inventory score ≥4	1125	11.5	13.0	12.3	13.7	714	14.4	16.0	14.9	17.0	411	8.5	10.0	9.1	10.9
Missing	790	8.1				322	6.5				468	9.7			
**Depression (accumulated)**															
Malaise inventory score ≥4 (never)	5681	58.0	79.7	78.2	81.2	2805	56.5	74.0	72.2	75.8	2876	59.7	85.6	84.0	87.2
Malaise inventory score ≥4 (once)	860	8.8	13.3	12.4	14.1	550	11.1	16.1	14.9	17.2	310	6.4	10.4	9.3	11.5
Malaise inventory score ≥4 (twice)	286	2.9	4.6	3.9	5.4	208	4.2	6.3	5.3	7.3	78	1.6	2.9	2.1	3.6
Malaise inventory score ≥4 (thrice)	124	1.3	2.4	2.0	2.9	99	2.0	3.6	2.9	4.3	25	0.5	1.1	0.7	1.5
Missing	2838	29.0				1306	26.3				1532	31.8			
**Marital Status (Age 50)**															
Married/Civil partnership	6744	68.9	68.9	68.0	69.8	3384	68.1	68.1	66.8	69.4	3360	69.7	69.7	68.4	71.0
Single, Never married/Civil partnership	1064	10.9	10.9	10.3	11.5	467	9.4	9.4	8.6	10.2	597	12.4	12.4	11.5	13.3
Separated/Divorced	1840	18.8	18.8	18.0	19.6	1015	20.4	20.4	19.3	21.6	825	17.1	17.1	16.0	18.2
Widowed	140	1.4	1.4	1.2	1.7	101	2.0	2.0	1.6	2.4	39	0.8	0.8	0.6	1.1
Missing	1	0.0				1	0.0				0	0.0			
**Employment Status (Age 50)**															
Employed	8256	84.3	84.4	83.6	85.1	4007	80.7	80.7	79.6	81.8	4249	88.1	88.2	87.2	89.1
Not employed	1527	15.6	15.6	14.9	16.4	957	19.3	19.3	18.2	20.4	570	11.8	11.8	10.9	12.8
Missing	6	0.1				4	0.1				2	0.0			
**Highest Qualification (Age 50)**															
Level 4/Degree or higher	2360	24.1	24.1	23.3	25.0	1231	24.8	24.8	23.6	26.0	1129	23.42	23.4	22.2	24.6
A Level/equivalent NVQ3	774	7.9	7.9	7.4	8.4	390	7.9	7.9	7.1	8.6	384	7.97	8.0	7.2	8.7
O Level/equivalent NVQ2	3328	34.0	34.0	33.1	35.0	1809	36.4	36.4	35.1	37.8	1519	31.51	31.5	30.2	32.8
CSE 2-5/equivalent NVQ1	1411	14.4	14.4	13.7	15.1	663	13.4	13.4	12.4	14.3	748	15.52	15.5	14.5	16.6
No qualification	1910	19.5	19.5	18.8	20.3	873	17.6	17.6	16.5	18.7	1037	21.51	21.5	20.4	22.7
Missing	6	0.1				2	0.0				4	0.08			
**Subjective Financial Status (Age 50)**															
Living comfortably	3827	39.1	39.1	38.2	40.1	1967	39.6	39.6	38.3	41.0	1860	38.6	38.6	37.3	40.0
Doing all right	2992	30.6	30.6	29.7	31.5	1528	30.8	30.8	29.5	32.1	1464	30.4	30.5	29.2	31.8
Just about getting by	2112	21.6	21.7	20.9	22.5	1033	20.8	20.9	19.7	22.0	1079	22.4	22.5	21.3	23.7
Finding it quite difficult	566	5.8	5.8	5.4	6.3	302	6.1	6.1	5.5	6.8	264	5.5	5.5	4.9	6.2
Finding it very difficult	264	2.7	2.7	2.4	3.0	126	2.5	2.6	2.1	3.0	138	2.9	2.9	2.4	3.4
Missing	28	0.3				12	0.2				16	0.3			
**Social Class of Father’s Occupation**															
Non-manual	2845	29.1	34.3	33.3	35.3	1441	29.0	34.1	32.7	35.5	1404.0	29.1	34.6	33.2	36.0
Manual	5479	56.0	65.7	64.7	66.7	2793	56.2	65.9	64.5	67.3	2686.0	55.7	65.4	64.0	66.8
Missing	1465	15.0				734	14.8				731	15.16			
**Childhood Disadvantage**															
None recorded	6838	69.9	70.3	69.4	71.2	3424	68.9	69.4	68.1	70.7	3414	70.8	71.3	70.0	72.5
One item recorded	2119	21.7	21.8	20.9	22.6	1118	22.5	22.6	21.5	23.8	1001	20.8	20.9	19.7	22.0
Two items recorded	522	5.3	5.4	4.9	5.8	268	5.4	5.4	4.8	6.1	254	5.3	5.3	4.7	5.9
Three items recorded	198	2.0	2.0	1.8	2.3	100	2.0	2.0	1.6	2.4	98	2.0	2.0	1.6	2.4
Four items recorded	51	0.5	0.5	0.4	0.7	26	0.5	0.5	0.3	0.7	25	0.5	0.5	0.3	0.7
Missing	61	0.6				32	0.6				29	0.6			
**Parental Discord at Age 7**															
No	7094	72.5	95.1	94.6	95.5	3623	72.9	95.1	94.5	95.8	3471	72.0	95.0	94.3	95.7
Yes	382	3.9	4.9	4.5	5.4	192	3.9	4.9	4.2	5.5	190	3.9	5.0	4.3	5.7
Missing	2313	23.6				1153	23.2				1160	24.1			
**Parents Separated or Died by Age 7**															
No	8563	87.5	95.6	95.2	96.0	4347	87.5	95.3	94.6	95.9	4216	87.5	95.9	95.3	96.5
Yes	395	4.0	4.4	4.0	4.8	217	4.4	4.7	4.1	5.4	178	3.7	4.1	3.5	4.7
Missing	831	8.5				404	8.1				427	8.9			
**BSAG (Age 7 and 11 Mean Score)**															
1st Quartile	2530	25.9	32.2	31.2	33.3	1550.0	31.2	38.9	37.4	40.4	980	20.3	25.3	23.9	26.7
2nd Quartile	1881	19.2	24.1	23.1	25.0	991.0	20.0	25.0	23.7	26.3	890	18.5	23.2	21.9	24.5
3rd Quartile	1776	18.1	23.1	22.2	24.1	825.0	16.6	21.1	19.7	22.4	951	19.7	25.2	23.9	26.6
4th Quartile	1564	16.0	20.6	19.7	21.5	581.0	11.7	15.1	14.0	16.1	983	20.4	26.3	25.0	27.6
Missing	2038	20.8				1021.0	20.6				1017	21.1			
**Smoking (Age 23)**															
Non-smoker	5136	52.5	60.9	59.9	61.9	2654	53.4	61.1	59.8	62.5	2482	51.5	60.7	59.3	62.2
Current or past smoker	3173	32.4	39.1	38.1	40.1	1621	32.6	38.9	37.5	40.2	1552	32.2	39.3	37.8	40.7
Missing	1480	15.1				693	14.0				787	16.3			
**Alcohol Consumption (Age 23)**															
≤14 units per week	3924	40.1	61.6	60.4	62.7	2452	49.4	80.6	79.2	81.9	1472	30.53	42.0	40.3	43.6
>14 units per week	2883	29.5	38.4	37.3	39.6	688	13.9	19.4	18.1	20.8	2195	45.53	58.0	56.4	59.7
Missing	2982	30.5				1828	36.8				1154	23.94			
**Physical Activity (Age 16)**															
Active	3951	40.4	52.9	51.8	54.0	1827	36.8	0.8	46.0	49.1	2124	44.1	58.4	56.9	59.9
Inactive	3487	35.6	47.1	46.0	48.2	1995	40.2	52.5	50.9	54.0	1492	31.0	41.6	40.1	43.1
Missing	2351	24.0				1146	23.1				1205	25.0			

Note: LCI = Lower Confidence Interval; UCI = Upper Confidence Interval.

**Table 2 jcm-12-05568-t002:** Associations between depression, life course exposures and frailty at age 50 (models adjusted for all covariates).

	Depression at Age 23				Depression at Age 33				Depression at Age 42				Accumulated Depression Score			
Independent Variables	OR	LCI	UCI	*p*-Value	OR	LCI	UCI	*p*-Value	OR	LCI	UCI	*p*-Value	OR	LCI	UCI	*p*-Value
**Depression**																
Malaise inventory score <4	Ref				Ref				Ref							
Malaise inventory score ≥4	**1.76**	**1.47**	**2.10**	**<0.001**	**2.11**	**1.72**	**2.60**	**<0.001**	**2.15**	**1.87**	**2.48**	**<0.001**				
**Depression (Accumulated)**																
Malaise inventory score ≥4 (never)													Ref			
Malaise inventory score ≥4 (once)													**1.92**	**1.65**	**2.23**	**<0.001**
Malaise inventory score ≥4 (twice)													**2.33**	**1.85**	**2.94**	**<0.001**
Malaise inventory score ≥4 (thrice)													**2.95**	**2.11**	**4.11**	**<0.001**
**Sex**																
Female													Ref			
Male	1.12	1.00	1.25	0.054	1.10	0.99	1.23	0.084	1.11	0.99	1.24	0.079	**1.17**	**1.05**	**1.32**	**0.005**
**Marital Status (Age 50)**																
Married/Civil partnership	Ref				Ref				Ref				Ref			
Single, Never married/Civil partnership	1.14	0.98	1.34	0.086	1.13	0.97	1.32	0.120	1.13	0.97	1.32	0.127	1.12	0.96	1.31	0.162
Separated/Divorced	0.89	0.79	1.01	0.065	**0.88**	**0.78**	**1.00**	**0.045**	**0.88**	**0.77**	**0.99**	**0.040**	**0.87**	**0.77**	**0.99**	**0.029**
Widowed	1.04	0.71	1.53	0.848	1.02	0.69	1.50	0.931	1.03	0.70	1.52	0.896	1.02	0.69	1.51	0.908
**Employment Status (Age 50)**																
Employed	Ref				Ref				Ref				Ref			
Not employed	**2.80**	**2.46**	**3.17**	**<0.001**	**2.74**	**2.41**	**3.11**	**<0.001**	**2.64**	**2.33**	**3.00**	**<0.001**	**2.65**	**2.33**	**3.01**	**<0.001**
**Highest Qualification (Age 50)**																
Level 4/Degree or higher	Ref				Ref				Ref				Ref			
A Level/equivalent NVQ3	1.02	0.82	1.27	0.831	1.04	0.83	1.29	0.746	1.04	0.84	1.30	0.715	1.04	0.83	1.29	0.738
O Level/equivalent NVQ2	**1.54**	**1.34**	**1.77**	**<0.001**	**1.54**	**1.34**	**1.77**	**<0.001**	**1.55**	**1.35**	**1.78**	**<0.001**	**1.54**	**1.34**	**1.77**	**<0.001**
CSE 2-5/equivalent NVQ1	**2.03**	**1.72**	**2.39**	**<0.001**	**2.04**	**1.72**	**2.40**	**<0.001**	**2.07**	**1.75**	**2.45**	**<0.001**	**2.03**	**1.72**	**2.40**	**<0.001**
No qualification	**2.45**	**2.09**	**2.88**	**<0.001**	**2.44**	**2.08**	**2.87**	**<0.001**	**2.49**	**2.12**	**2.93**	**<0.001**	**2.40**	**2.03**	**2.82**	**<0.001**
**Subjective Financial Position (Age 50)**																
Living comfortably	Ref				Ref				Ref				Ref			
Doing all right	**1.21**	**1.07**	**1.36**	**0.002**	**1.21**	**1.08**	**1.36**	**0.002**	**1.20**	**1.07**	**1.35**	**0.002**	**1.19**	**1.06**	**1.34**	**0.003**
Just about getting by	**1.65**	**1.45**	**1.87**	**<0.001**	**1.62**	**1.43**	**1.84**	**<0.001**	**1.60**	**1.41**	**1.82**	**<0.001**	**1.59**	**1.40**	**1.80**	**<0.001**
Finding it quite difficult	**2.20**	**1.80**	**2.69**	**<0.001**	**2.23**	**1.83**	**2.73**	**<0.001**	**2.14**	**1.75**	**2.61**	**<0.001**	**2.12**	**1.74**	**2.60**	**<0.001**
Finding it very difficult	**2.91**	**2.18**	**3.90**	**<0.001**	**2.96**	**2.21**	**3.96**	**<0.001**	**2.78**	**2.07**	**3.73**	**<0.001**	**2.69**	**2.00**	**3.62**	**<0.001**
**Social Class of the Father’s Occupation (at Birth)**																
Non-manual	Ref				Ref				Ref				Ref			
Manual	**1.30**	**1.15**	**1.46**	**<0.001**	**1.30**	**1.15**	**1.46**	**<0.001**	**1.30**	**1.15**	**1.46**	**<0.001**	**1.30**	**1.16**	**1.47**	**<0.001**
**Childhood Disadvantage**																
None recorded	Ref				Ref				Ref				Ref			
One item recorded	1.00	0.89	1.13	0.961	1.00	0.89	1.13	0.950	1.00	0.89	1.13	0.982	0.99	0.88	1.11	0.839
Two items recorded	1.16	0.95	1.42	0.155	1.16	0.95	1.43	0.145	1.16	0.95	1.43	0.147	1.15	0.94	1.41	0.185
Three items recorded	1.38	1.00	1.91	0.050	1.33	0.96	1.84	0.087	1.33	0.96	1.85	0.083	1.35	0.97	1.87	0.071
Four items recorded	0.97	0.53	1.78	0.920	0.96	0.52	1.77	0.903	0.93	0.51	1.71	0.823	0.92	0.50	1.69	0.787
**Parental Discord at Age 7**																
No	Ref				Ref				Ref				Ref			
Yes	0.99	0.77	1.27	0.939	0.98	0.76	1.27	0.894	1.01	0.78	1.31	0.933	0.96	0.74	1.24	0.752
**Parents Divorced, Separated or Died by Age 7**																
No	Ref				Ref				Ref							
Yes	**1.40**	**1.11**	**1.77**	**0.005**	**1.40**	**1.10**	**1.78**	**0.005**	**1.37**	**1.08**	**1.73**	**0.010**	**1.39**	**1.09**	**1.76**	**0.007**
**BSAG (Age 7 and 11 Mean Score)**																
1st Quartile	Ref				Ref				Ref				Ref			
2nd Quartile	1.15	1.00	1.33	0.053	**1.16**	**1.01**	**1.34**	**0.042**	**1.16**	**1.00**	**1.33**	**0.048**	1.15	0.99	1.32	0.063
3rd Quartile	**1.30**	**1.12**	**1.51**	**0.001**	**1.30**	**1.12**	**1.51**	**<0.001**	**1.30**	**1.12**	**1.51**	**0.001**	**1.28**	**1.10**	**1.48**	**<0.001**
4th Quartile	**1.48**	**1.26**	**1.73**	**<0.001**	**1.50**	**1.28**	**1.75**	**<0.001**	**1.50**	**1.28**	**1.75**	**<0.001**	**1.44**	**1.23**	**1.69**	**<0.001**
**Smoking (Age 23)**																
Non-smoker	Ref				Ref				Ref				Ref			
Current or past smoker	**1.14**	**1.03**	**1.27**	**0.010**	**1.15**	**1.04**	**1.28**	**0.008**	**1.16**	**1.04**	**1.28**	**0.007**	**1.13**	**1.02**	**1.26**	**0.019**
**Alcohol Consumption (Age 23)**																
≤14 units per week	Ref				Ref				Ref				Ref			
>14 units per week	**0.83**	**0.74**	**0.95**	**0.005**	**0.84**	**0.74**	**0.95**	**0.007**	**0.83**	**0.73**	**0.95**	**0.006**	**0.84**	**0.74**	**0.95**	**0.006**
**Physical Activity (Age 16)**																
Active	Ref				Ref				Ref				Ref			
Inactive	1.09	0.98	1.22	0.120	1.09	0.98	1.22	0.118	1.08	0.96	1.21	0.206	1.07	0.96	1.20	0.240

Note: OR = Odds Ratio; LCI = Lower Confidence Interval; UCI = Upper Confidence Interval; Values in bold indicate a *p*-value below 0.05.

## Data Availability

All NCDS data used in this study are freely available for download from the UK Data Service: https://ukdataservice.ac.uk/.

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
