# Peer review of "Depression Earlier on in Life Predicts Frailty at 50 Years: Evidence from the 1958 British Birth Cohort Study"

_jcm, 2023, doi:10.3390/jcm12175568_

Round 1

Reviewer 1 Report

I really liked your work, congratulations for the effort. I find it really interesting and I hope you will continue in this line of research and in the future we will be able to see your work on depression-anxiety, chronic inflammation and disease, I think it is really interesting and to be able to address disease prevention management in the population.

I would like to know if any of the patients included in the work had a serious situation in childhood, not separation or divorce of parents or orphanhood, but if any of them could have suffered a serious illness and this conditioned them to have a more depressive and greater mood fragility. I also see that you have not defined a conclusion at the end of the work, but that it is as indicated in the discussion but it would be good to indicate a sentence, a small paragraph as the final conclusion. Again, congratulations on the job.

Author Response

Reviewer: 1

Comments and Suggestions for Authors

I really liked your work, congratulations for the effort. I find it really interesting and I hope you will continue in this line of research and in the future we will be able to see your work on depression-anxiety, chronic inflammation and disease, I think it is really interesting and to be able to address disease prevention management in the population.

Many thanks for your interest and encouraging comments.

I would like to know if any of the patients included in the work had a serious situation in childhood, not separation or divorce of parents or orphanhood, but if any of them could have suffered a serious illness and this conditioned them to have a more depressive and greater mood fragility.

Thank you for this comment. A serious illness in childhood is indeed a potential confounding factor in the relationship between depression and frailty. We investigated the impact of this potential confounder by conducting sensitivity analyses excluding participants with a physical handicap or disabling condition at age 7 or a longstanding illness or disability at age 23. In these sensitivity analyses all associations remained with minimal changes in effect size. These findings are reported in lines 234-236 of the manuscript.

I also see that you have not defined a conclusion at the end of the work, but that it is as indicated in the discussion but it would be good to indicate a sentence, a small paragraph as the final conclusion. Again, congratulations on the job.

Thank you. We have added a concluding paragraph to the manuscript.

Reviewer 2 Report

The study highlights an important topic about depression and frailty measured at different life stages, please find the comments below

·         The abstract is well written and informative

·        Depression, the main exposure, was measured using the 9-item Malaise Inventory, any reasons why this scale was used? For example, the PHQ-9 is also a validated and a reliable scale

·        The excellent sample size, the longitudinal design and the richness of the covariates are strengths in the study

·        The outcome variable and the exposure were treated as categorical variables; therefore, a multivariate logistic regression was carried out. Did the authors try to treat these variables as scale variables and therefore perform a linear regression model instead?

·        The data analysis reads: “Associations between frailty at age 50 and depression at ages 23, 33, and 42 were 195 estimated using multiple logistic regression models”, did the authors perform a preliminary univariate analysis to know the potential factors to be included in the final multivariate regression model?  Please make this part clear

·        Physical health in this age group namely hypertension, diabetes, high cholesterol levels and their medications are considered potential confounding factors with frailty, the authors did not take them into consideration, was this intentional or the information were lacking?

·        Antidepressants were not studied, are there any data about the use of antidepressants? However, they were mentioned in the discussion

·        The manuscript needs  more accurate concluding remarks

Author Response

Reviewer 2:

Comments and Suggestions for Authors

The study highlights an important topic about depression and frailty measured at different life stages, please find the comments below:

abstract is well written and informative.

Many thanks for your interest and encouraging comments.

Depression, the main exposure, was measured using the 9-item Malaise Inventory, any reasons why this scale was used? For example, the PHQ-9 is also a validated and a reliable scale.

Thank you for this question. We used the 9-item Malaise inventory as it was the most appropriate measure of depression available at multiple waves across the life course in the NCDS dataset. The measurement invariance properties identified in previous studies were considered to be a strength of this measure. Furthermore, we selected the nine-item version to ensure that items in the exposure measure did not cross over with items in the outcome measure. Further research, using datasets with alternative measures of depression available will be important in identifying whether the associations we have identified are consistent when using different measurement tools. We have added lines 333-335 to the manuscript to highlight this point.

The excellent sample size, the longitudinal design and the richness of the covariates are strengths in the study.

Thank you. These are indeed strengths of the NCDS dataset, and we acknowledge the contributions of so many people who have worked to make this rich dataset available since the NCDS inception in 1958.

The outcome variable and the exposure were treated as categorical variables; therefore, a multivariate logistic regression was carried out. Did the authors try to treat these variables as scale variables and therefore perform a linear regression model instead?

Thank you for this comment. We used a categorical variable for our frailty outcome as the cut off point off 0.25 has been used most commonly in frailty research and is recommended by researchers who have studied the validity of this measure. The use of a continuous measure is much less common in frailty research. This is largely due to the evidence showing that the binary measure is strongly associated with outcomes such as mortality, hospitalisation and the need for full time care. The interpretation of a continuous frailty measure is less clear in relation to these subsequent outcomes.

We used a categorical exposure measure for similar reasons. The cut off point of >=4 has been consistently used in previous studies to represent depressive symptoms being present. The binary measure is therefore more clearly interpreted than a continuous measure and by using this cut off point our study can be more easily compared to previous research.

The data analysis reads: “Associations between frailty at age 50 and depression at ages 23, 33, and 42 were estimated using multiple logistic regression models”, did the authors perform a preliminary univariate analysis to know the potential factors to be included in the final multivariate regression model?  Please make this part clear.

Thank you for this comment. We did conduct preliminary univariate analyses, however, our main rationale for the inclusion of independent variable was their theorised relationship with both the exposure and the outcome and evidence from previous studies indicating their likely association with the exposure and outcome. We have added lines 155-157 to the manuscript to clarify this point.

Physical health in this age group namely hypertension, diabetes, high cholesterol levels and their medications are considered potential confounding factors with frailty, the authors did not take them into consideration, was this intentional or the information were lacking?

Thank you for this comment. Physical health conditions such as hypertension, diabetes, high cholesterol levels are eligible to be included as items in a frailty index and could therefore not be considered as confounding factors. Indeed, hypertension and diabetes were included in our frailty index (please see supplementary file 1.)

Antidepressants were not studied, are there any data about the use of antidepressants? However, they were mentioned in the discussion.

Thank you for this comment. Unfortunately, detailed information about antidepressant use is not available in the NCDS dataset. Future research in the role of antidepressant use in the relationship between depression and frailty will make a valuable contribution to this area. We have discussed the potential role of antidepressants in lines 297-302.

The manuscript needs more accurate concluding remarks.

Thank you. We have added a concluding paragraph to the manuscript.